# Modeling of Three-Way Catalyst Dynamics for a Compressed Natural Gas Engine during Lean–Rich Transitions

**Dario Di Maio** [1,2,*], **Carlo Beatrice** [1,*], **Valentina Fraioli** [1], **Pierpaolo Napolitano** [1], **Stefano Golini** [3] **and Francesco Giovanni Rutigliano** [3]

1   CNR – Istituto Motori, Via G. Marconi, 4, 80125 Naples, Italy; v.fraioli@im.cnr.it (V.F.);
    p.napolitano@im.cnr.it (P.N.)
2   University of Naples "Parthenope", Centro Direzionale Isola C4, 80143 Naples, Italy
3   FPT Industrial SpA, Via Puglia, 15, 10156 Turin, Italy; stefano.golini@cnhind.com (S.G.);
    francesco.rutigliano@cnhind.com (F.G.R.)
*   Correspondence: d.dimaio@im.cnr.it (D.D.M.); c.beatrice@im.cnr.it (C.B.);
    Tel.: +39-081-7177169 (D.D.M.); +39-081-7177186 (C.B.)

**Abstract:** The main objective of the present research activity was to investigate the effect of very fast composition transitions of the engine exhaust typical in real-world driving operating conditions, as fuel cutoff phases or engine misfire, on the aftertreatment devices, which are generally very sensitive to these changes. This phenomenon is particularly evident when dealing with engines powered by natural gas, which requires the use of a three-way catalyst (TWC). Indeed, some deviations from the stoichiometric lambda value can interfere with the catalytic converter efficiency. In this work, a numerical "quasi-steady" model was developed to simulate the chemical and transport phenomena of a specific TWC for a compressed natural gas (CNG) heavy-duty engine. A dedicated experimental campaign was performed in order to evaluate the catalyst response to a defined λ variation pattern of the engine exhaust stream, thus providing the data necessary for the numerical model validation. Tests were carried out to reproduce oxygen storage phenomena that make catalyst behavior different from the classic steady-state operating conditions. A surface reaction kinetic mechanism concerning $CH_4$, CO, $H_2$, oxidation and NO reduction has been appropriately calibrated at different λ values with a step-by-step procedure, both in steady-state conditions of the engine work plan and during transient conditions, through cyclical and consecutive transitions of variable frequency between rich and lean phases. The activity also includes a proper calibration of the reactions involving cerium inside the catalyst in order to reproduce oxygen storage and release dynamics. Sensitivity analysis and continuous control of the reaction rate allowed evaluating the impact of each of them on the exhaust composition in several operating conditions. The proposed model predicts tailpipe conversion/formation of the main chemical species, starting from experimental engine-out data, and provides a useful tool to evaluate the catalyst's performance.

**Keywords:** TWC (three-way catalyst); natural gas; kinetic scheme; aftertreatment; modeling; emission; internal combustion engines

---

## 1. Introduction

Automotive industries are dealing with significant advancements in pollutant control strategies to comply with increasingly stringent emission standards [1]. Especially in heavy-duty truck and bus engines, compressed natural gas (CNG) has become more and more attractive as alternative fuel in

terms of emissions and performance in comparison to traditional fuels, reducing adverse health effects and social costs of air pollution [2].

For spark ignition (SI) stoichiometric CNG engines, the most suitable pollutant abatement system is the three-way catalytic converter (TWC). Similar to SI gasoline engines, this device permits control of $NO_x$, unburned methane ($CH_4$), and other pollutant emissions (CO, NMHC). The simultaneous conversion of these species is possible exclusively in a very tiny range of inlet stream composition around stoichiometric conditions [3]. Compared to gasoline engines, this optimum operating point is further reduced because of a sudden drop in $NO_x$ conversion efficiency, as soon as a slightly lean λ value is achieved, and a non-complete conversion of total hydrocarbons (THCs) both in lean and in rich conditions [4].

Previous studies have widely demonstrated the differences in conversion efficiencies between a steady-state test and dynamic conditions [5,6]. Some species of the washcoat layer may oxidize or reduce depending on the exhaust gas composition. For TWCs, the most significant of these components is cerium, which acts a stabilizer and a medium for oxygen storage. These effects have a considerable impact on the λ value inside the catalyst, which must be appropriately investigated in order to better manage TWC behavior. Under real-world driving conditions, several deviations of the air-to-fuel ratio (AFR) from the stoichiometric value take place because of fuel cutoff phases, engine misfire, and response lag of the fuel control system [7].

In this scenario, the numerous tasks and the mentioned working issues typical of this aftertreatment device have required, since its introduction, a development of reliable and predictive numerical models capable of analyzing specific operating conditions.

Modeling approaches are generally categorized as 0D, 1D, and 3D, with increasing accuracy and complexity levels. A zero-dimensional approach is solely used for steady-state conditions, as only the mean exhaust gas mass flow is considered, thus reducing the reacting device to an element that performs the chemical conversion according to kinetic parameters given as input. More complex is the 3D approach, used when the characterization of flow distribution inside the reactor is required. The accuracy is certainly improved because it permits to identify radial diffusion effects of the flow inside the catalyst; the real limitation is represented by high computational efforts, which do not allow an investigation in a very wide range of operating conditions, such as the variables of a dynamic cycle, in times compatible with project targets. A good compromise is represented by 1D models, which allow to simulate a thermo-fluid dynamic behavior of the whole exhaust system during both steady-state and transient conditions with a reduced computational effort [8].

Calibration of the kinetic reaction model inside the catalyst is certainly one of the most challenging topics. Several studies are available in literature, mostly concerning traditional gasoline spark ignition (SI) engines [9–11].

An interesting modeling approach, based on 104 reaction steps, of a TWC kinetic scheme with an exhaust mixture from natural gas-fueled engines was proposed by Zeng et al. However, in this study, only steady-state (SS) conditions were analyzed; thus, oxygen storage phenomena and perturbations in AFR were not considered [12]. Very little information is available in the literature on the specific features of TWC systems applied to natural gas vehicles. In this respect, an important contribution is provided by Tsinoglou et al. through a comparison between honeycomb and ceramic foam catalysts [13].

With a favorable ratio between accuracy and calculation time, in the present work a "quasi-steady" model, equipped with comprehensive oxygen storage and release submodels, is setup to analyze the effect of cyclical perturbations in the exhaust gas of a NG heavy-duty engine on TWC efficiency in different load conditions. Catalyst performances under fast transient AFR dynamics, from lean to very rich conditions, are investigated.

## 2. Experimental Investigation

### 2.1. Engine and Catalyst Characterization

The experimental activities were conducted on a 6-cylinder heavy duty natural gas production engine, with a combustion system design compliant with Euro VI regulations. The port fuel injection (PFI) NG injectors were fed by a separate NG low-pressure line operating at a pressure of about 10 barG. NG consumption was measured by means of an Emerson Coriolis effect device; the air flow rate was measured by means of an air mass flow meter. The experimental layout is reported in Figure 1, while Table 1 describes the main characteristics of the engine. The chemical composition of the adopted fuel is summarized in Table 2.

**Table 1.** Main features of the natural gas 6-cylidnder engine.

| | |
|---|---|
| Displaced volume | 12.8 L |
| Stroke | 150 mm |
| Bore | 135 mm |
| Compression ratio | 12:1 |
| Number of Valves | 4 |
| Rated Power | 338 kW @ 2000 rpm |
| Torque | 2000 Nm @ 1100–1620 rpm |
| PFI Injector | Natural gas |

**Table 2.** CNG fuel mixture used during experimental tests.

| Fuel Composition | | |
|---|---|---|
| Methane | $CH_4$ | 84.78% |
| Ethane | $C_2H_6$ | 8.88% |
| Propane | $C_3H_8$ | 1.88% |
| N-Butane | $C_4H_{10}$ | 0.50% |
| N-Pentane | $C_5H_{12}$ | 0.08% |
| N-Hexane | $C_6H_{14}$ | 0.04% |
| Nitrogen | $N_2$ | 1.90% |
| Carbon Dioxide | $CO_2$ | 1.87% |
| Helium | $He$ | 0.07% |

All pollutant emissions were measured by AVL i60 devices: THC and $CH_4$ by a flame ionization detector (FID), $NO_x$ by a chemi-luminescence detector (CLD), CO and $CO_2$ by an infrared detector (IRD), $O_2$ by a paramagnetic detector (PMD), and $NH_3$ by an AVL LDD standalone ammonia analyzer.

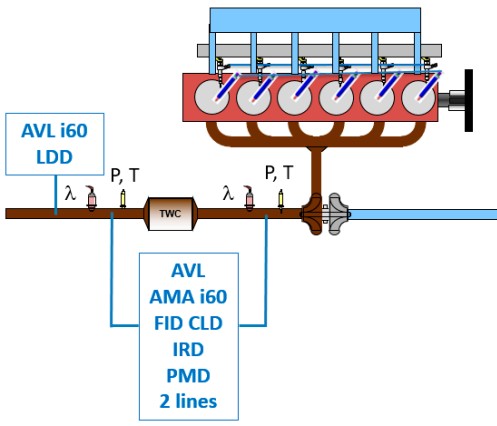

**Figure 1.** Experimental layout.

## 2.2. Test Bench Data

The experimental tests were performed in two ways. For each test, the AFR engine control unit (ECU) was disabled, and the AFR was superimposed by means of injection quantity at a fixed air mass flow. The experimental tests in SS conditions (15 engine operative points) were carried out with λ sweep from 0.90 to 1.10. However, the real value of AFR can be slightly different from the value derived from ECU, taking into account the variable measurement dynamics of the chemical species analyzers and the behavior of the installed λ sensors.

Several analyses have been carried out in order to identify a reference λ value to describe discrepancies in exhaust gas composition. In fact, the engine was equipped with two types of sensors that provided AFR measurements. The first is the Smart $NO_x$ Sensor (SNS) 120 by Continental™, consisting of a ceramic sensor element made of zirconia ($ZrO_2$) and an electronic control unit. The second is the Universal Lambda Sensor (ZFAS-U) by NGK™, made of two zirconia ($ZrO_2$) substrate elements, one is the $O_2$ pumping cell (Ip cell), the other is the $O_2$ detecting cell (Vs cell) heated by a ceramic heater, which is supplied by a very small current. The SNS sensor has a very rapid response to composition changes and is very precise at stoichiometric conditions, but it presents major uncertainties when the exhaust gas is in rich or lean conditions because of the linear correlation between oxygen concentration of residual gas and AFR. On the other hand, for the ZFAS-U sensor, very precise at stoichiometric conditions, even a slight variation in the exhaust gas composition typical of NG engines, can considerably affect the sensed AFR value, which leads to oscillating values. For these reasons, it is not reliable in transient operations but remains suitable for SS conditions.

λ values can also be calculated starting from the classic ratio between air and fuel flow intake rate and fuel and stoichiometric dosage. However, since the main constituents of the CNG mixture do not have a specific concentration throughout the duration of the test, but is included within a range of appropriate values, the value of $\alpha_{st}$ is changeable. Moreover, especially in low-load conditions, deviations and measurement uncertainties of air and fuel flow increase and interfere with the result of the ratio.

Therefore, the most reliable method to calculate λ for SS conditions was identified as an analytical relation present in regulation 49 [14] from test best bench data:

$$\lambda_i = \frac{\left(100 - \frac{c_{CO_d} \cdot 10^{-4}}{2} - c_{HC_w} \cdot 10^{-4}\right) + \left(\frac{\alpha}{4} \cdot \frac{1 - \frac{2 \cdot c_{CO_d} \cdot 10^{-4}}{3.5 \cdot c_{CO_{2d}}}}{1 + \frac{c_{CO_d} \cdot 10^{-4}}{3.5 \cdot c_{CO_{2d}}}} - \frac{\varepsilon}{2} - \frac{\delta}{2}\right) * \left(c_{CO_{2d}} + c_{CO_d} \cdot 10^{-4}\right)}{4.764 * \left(1 + \frac{\alpha}{4} - \frac{\varepsilon}{2} + \gamma\right) * \left(c_{CO_{2d}} + c_{CO_d} \cdot 10^{-4} + c_{HC_w} \cdot 10^{-4}\right)}, \tag{1}$$

where:

$\lambda_i$ is the instantaneous "relative air-to-fuel ratio";
$c_{CO_{2d}}$ is the dry $CO_2$ concentration, in percentage;
$c_{CO_d}$ is the dry CO concentration, ppm;
$c_{HC_w}$ is the wet HC concentration, ppm;
$\alpha$ is the molar hydrogen ratio (H/C);
$\beta$ is the molar carbon ratio (C/C);
$\gamma$ is the molar sulfur ratio (S/C);
$\delta$ is the molar nitrogen ratio (N/C); and
$\varepsilon$ is the molar oxygen ratio (O/C);

The usage of Equation (1) for the λ calculation allows to disregard the use of sensors and their mentioned inaccuracies, as this calculation is directly based on the species' concentrations actually measured in the engine exhaust.

Not surprisingly, despite the assumptions that will be fully described in the next paragraph, this value is completely in line with what is calculated by the model. In fact, the simulation tool uses an atomic formulation, available in the GT-Suite manual, for calculating λ value that is precisely correlated to the chemical species present in exhaust (Equation (2)). It is derived as a ratio between total oxygen atoms and total oxygen atoms required to oxidize all carbon, hydrogen, and sulfur atoms.

$$\lambda = \frac{O}{2 \cdot (C + S) + \frac{H}{2}}. \tag{2}$$

Figure 2 summarizes all these considerations by highlighting the trend of the λ measurements and the discrepancies in the use of a formulation rather than another. Some differences in λ calculations are found even if the operating condition is the same. Given the high sensitivity of TWC in the stoichiometric vicinity, in particular for CNG engines, in order to evaluate the goodness of a reaction's kinetic scheme it is necessary to observe the measurements of single species that will finally give the correct λ value.

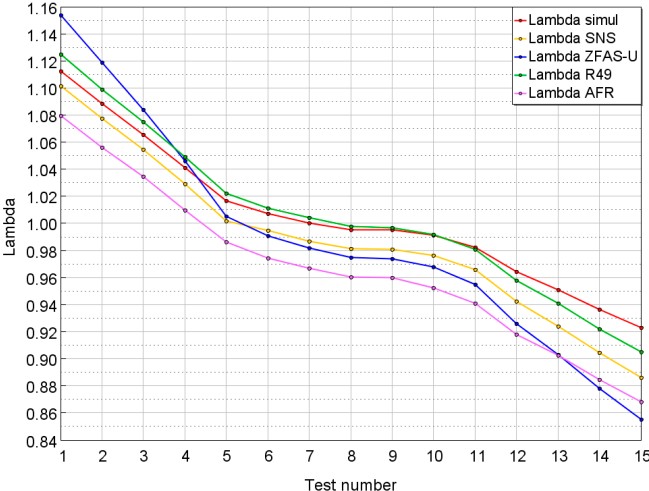

**Figure 2.** Comparison of λ values calculated with different formulations at 1900 RPM (100%) load. On the *y*-axis, the average value of λ is shown for each of the 15 conditions tested. In red: λ from GT-Suite; in yellow: measured λ from the Smart $NO_x$ Sensor (SNS); in blue: measured λ from the Universal Lambda Sensor (ZFAS-U); in green: λ derived from the R49 regulation; in pink: λ from air/fuel ratio and stoichiometric dosage.

Finally, hydrogen concentration values were estimated. Such a hypothesis, despite λ values not being significantly affected, can certainly influence the conversion of pollutants, especially $NO_x$, and remains one of the points to be explored in further studies.

The experimental tests in dynamic conditions were carried out with a schematic pattern of the λ target reported in Figure 3. Usually, in the evaluation of the oxygen storage phenomena, and in the characterization of the catalyst's behavior changing in the transitions from rich to lean and vice versa, wide duration tests, similar to SS, were used for AFR scans [6], or some alternative, similar conditions were investigated with the introduction of a synthetic gas [15]. The innovative design of the experiments provides three consecutive transitions made through an AFR control system that allows the exploration of the emissions of a real engine during these rather complex phases. As shown in Figure 3, the target value of λ varied between 0.90 in rich conditions and 1.10 in lean conditions. The maximum duration of a single step was set equal to 10 s, as in this timeframe the exhaust gas, reaching a stationary composition, fully oxidized the cerium contained within the catalyst. On the contrary, the highest achievable frequency was set at 1 Hz. Below this value, the catalytic converter cannot follow the input dynamics, giving rise to an unaffected efficiency with faster oscillations. In fact, especially

at low engine loads, the presence of an empty volume upstream from the active catalytic zone can dampen the temporal evolution of the species' concentrations.

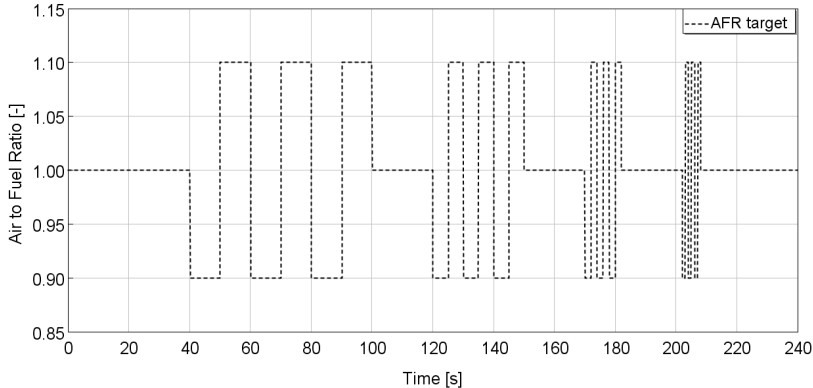

**Figure 3.** λ target in the dynamic experimental test.

The concentrations of the main exhaust gas components were measured before and after TWC. These values were determined by analyzing a fully dried sample stream for $CO_2$, CO, $O_2$, and $NO_x$ and a fully wet stream with a flame ionization detector (FID) for unburned hydrocarbons. Based on these experimental data, water concentration was calculated according the following formula:

$$\widetilde{x}_{H_2O} = \frac{m}{2n} \frac{\widetilde{x}^*_{CO} + \widetilde{x}^*_{CO_2}}{\left[1 + \widetilde{x}^*_{CO}/\left(K\,\widetilde{x}^*_{CO_2}\right) + (m/2n)\left(\widetilde{x}^*_{CO} + \widetilde{x}^*_{CO_2}\right)\right]}. \tag{3}$$

In the formula represented by Equation (3), K is a constant equal to 3.65, while m and n are typical values obtainable from the global chemical formula $C_nH_mO_r$ representing the employed NG mixture reported in Table 2, and $\widetilde{x}^*$ denotes the dry mole fraction of the species in subscript [16].

## 3. Assumptions

The main hypotheses commonly adopted for catalyst 1D models are still valid also for this application because, as mentioned in the introduction, the TWC "quasi-steady" mathematical model, here extended to CNG engines, was based on previous validated works for traditional gasoline engines. Along the catalytic converter, changes in potential and kinetic energy are neglected as well as heat losses to the surroundings. In order to analyze the TWC performance when exposed to real concentrations of the pollutants in the exhaust stream, it is necessary to measure, at the reactor inlet, the exhaust mass flow rate and temperature. Their values can be considered constant because these tests are carried out at a fixed load and engine speed. Very slight fluctuations in the measurements are to be attributed to the sensor's acquisition dynamics. A constant pressure was assumed along the system. Radial diffusion was not considered. Wet concentrations of main species present in exhaust gas as $CO_2$, CO, $H_2O$, NO, $NO_2$, $H_2$, $CH_4$, $C_3H_8$, $O_2$, and $N_2$ (evaluated as complement to unity of total mass flow) were imposed at the inlet of the TWC. As known, these categories of engines, given the high H/C ratio, produce a conspicuous quantity of hydrogen, which has an appreciable impact on catalyst reactions, especially on oxygen storage capacity (OSC). Since hydrogen concentration measurements in the exhaust gas were not available, an assumption had to be made starting from the empirical correlation from [17], which relates its values to CO concentrations for every AFR, as graphically represented in Figure 4. This characteristic has been extended to values with lean mixture conditions.

It is clear that hydrogen measurements both at the inlet and outlet of the TWC would allow a better calibration for the nitrogen and carbon oxide conversions. In addition, measurements could also be interesting in evaluating possible interactions with NO in the direct production of $NH_3$ and $N_2O$.

It is worth to mention that the residual oxygen concentration in exhaust gases, especially in stoichiometric and in rich conditions, represents a critical issue. Indeed, its concentration, measured by the analyzer, was approximately equal to 0.2%, even in the conditions at the lowest AFR. As a result, oxygen analyzer measurements turned out to be an important element for the simulation setup, influencing, as known, the oxidation/reduction of the pollutant species.

A $NO/NO_2$ ratio equal to 90/10 was used, in line with the assumptions generally made for traditional SI engines.

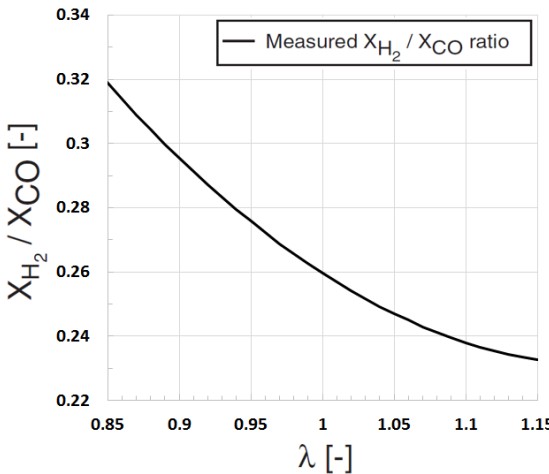

**Figure 4.** $H_2/CO$ ratio. Empirical correlation from Holder et al. [17].

## 4. Model Validation

The present paragraph briefly describes the procedure adopted to validate the kinetic conversion model of main gaseous pollutants and the oxygen storage phenomena. First of all, a default kinetic mechanism with a reaction scheme from [17] was the starting reference model. This mechanism, already implemented in a default model in GT-Suite and comprising 13 chemical species and 21 reactions, was specifically developed to summarize the most relevant reaction paths taking place in a TWC for exhaust streams from a gasoline engine. The governing equations are reported in the software manual and are also contained in [17].

Applying this initial model to simulations of an NG engine, it was not possible to obtain a correct prediction of the main pollutant species' concentrations at the tailpipe, even though the injection mode was similar to a traditional gasoline SI engine.

As an example, Figure 5 shows engine-out and tailpipe concentrations of some of the main converted species obtained through the default kinetic scheme [17]. For the sake of brevity, only the results of the most favorable conditions at high loads in SS were proposed. The wide deviations between the pink curve and the green curve are significant, demonstrating the inaccuracy of the starting model.

The following section describes the protocol adopted to perform TWC model calibrations, both in SS conditions and during lambda scans.

To this aim, a sensitivity analysis was performed for the pre-exponential factor **A** and the activation energy **E$_a$**, which characterize the rate expression of each reaction. The global reaction rates are generally expressed in this form [17]:

$$\omega = k \prod_k X_k^{\alpha_{kr}} \prod_m \Gamma_l \theta_m^{\beta_{mr}} / \prod_k F_j, \tag{4}$$

with the Arrhenius terms:

$$k \left( kmol/m^3 \right) = A T_s^\beta \exp[-E_a/RT_s]. \tag{5}$$

The calibration procedure began with steady-state cases, monitoring the rate of each reaction. The starting point was the deactivation of all reactions in order to obtain the same compositions in

engine-out and in tailpipe conditions. Then, an algorithm based on the order of activation of each reaction, dictated by the species that had higher concentrations in the exhaust, of the kinetic scheme were implemented. The platinum group metals (PGM) chemical reaction calibration procedure can be summarized in the following steps:

1. Deactivation of all reactions;
2. Activation of reactions with CO and $CO_2$ in reagents and products;
3. Run simulation and monitor reaction rates;
4. Further deactivation of all reactions except the one with the higher rate;
5. Parameterization of this reaction to get the result as close as possible to experiments;
6. Activation and calibration of each reaction based on the highest rate;
7. Repeat steps 2 to 6 for reactions with $CH_4$;
8. Check and recalibrate reactions with CO and $CO_2$;
9. Repeat steps 2 to 6 for reactions with NO; and
10. Check and recalibrate reactions with CO, $CO_2$, and $CH_4$.

Calibration of the kinetic scheme was carried out under the operating conditions 1900 RPM (100%) both in steady-state and transient conditions. The reaction model was then verified also at partial loads in order to validate its consistency.

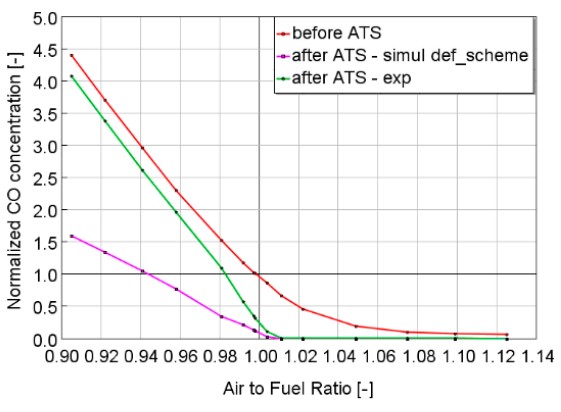 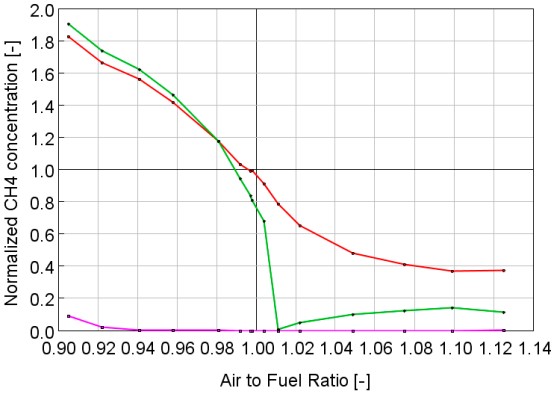

**Figure 5.** CO and $CH_4$ conversions with the uncalibrated (GT-Suite default) kinetic scheme from [17]. Gas Temp. $\approx 760\,°C$. (1900 RPM, 100% load).

*Oxygen Storage Submodel*

The same calibration protocol was used for the reactions involving cerium, which, as mentioned, is mainly responsible for the accumulation/release of oxygen (OSC). Cerium is normally present in high quantities in the washcoat (around 30% by weight, i.e., 1000 g/ft$^3$ or 35.31 kg/m$^3$). Cerium stabilizes the washcoat layer, enhances precious metal activity, and improves thermal resistance. Oxygen storage is due to cerium's ability to form 3- and 4-valent oxides [7]. The following reaction represents cerium oxidation and the corresponding oxygen capture:

$$\frac{1}{2}O_2 + Ce_2O_3 \rightarrow 2CeO_2$$

This reaction stands for the storage of an oxygen atom by increasing the cerium oxidation state. Because of the significant presence of carbon oxides and hydrogen in the exhaust gas, additional important pathways are:

$$CO + 2CeO_2 \leftrightarrow CO_2 + Ce_2O_3$$

$$H_2 + 2CeO_2 \leftrightarrow H_2O + Ce_2O_3$$

As demonstrated in previous works [7,11], cerium also interacts with nitrogen oxides, according to the following reaction:

$$NO + Ce_2O_3 \rightarrow \frac{1}{2}N_2 + 2CeO_2$$

which has been added to the initial kinetic scheme to better characterize the dynamics of TWC during fast transient $\lambda$ sweep. It also provides a mechanism for NO reduction under lean conditions, which can be particularly important under real-world driving conditions. Finally, the importance of this pathway was confirmed by the typical delay in the NO release at the TWC outlet, also detected in the present experimental activity.

## 5. Steady-State Results

This section shows the main results obtained through the kinetic scheme calibration. The following graphs show the concentration of significant chemical species, normalized with respect to stoichiometric emission values (to keep original equipment manufacturer (OEM) data confidential) and measured upstream and downstream from the catalyst. The figures in the present section report $CO_2$, $CO$, $CH_4$, and NO concentrations, combined with the AFR calculated through the R49 formula, as reported before. The univocal use of this formulation for the $\lambda$ calculation allowed not to consider the slight deviations that occurred in extremely rich or lean conditions, compared to the GT-Suite formulation, and to better appreciate the goodness of the kinetic scheme, focusing only on the concentration of each species. Looking at the *x*-axis of Figures 6–8, it is possible to notice that the $\lambda$ variation step was not uniformly spaced between lean and rich limits, but a higher number of datapoints were collected close to the stoichiometric value, in order to better describe the variation of the TWC efficiency in this specific critical range.

### 5.1. Carbon Oxides

The main reactions involving CO conversion in the TWC are essentially the direct oxidation via oxygen and the water gas shift (WGS) reported in the Appendix A as reaction 1 and reaction 6:

$$CO + \frac{1}{2}O_2 \rightarrow CO_2$$

$$CO + H_2O \rightarrow CO_2 + H_2$$

The first reaction has a high rate in lean and stoichiometric conditions, when a considerable amount of oxygen is present in the exhaust gas, while it has a weak influence in extremely rich conditions. The higher $H_2O$ concentration in the exhaust, compared to a traditional gasoline engine, makes WGS very influential among carbon oxide reactions. For this reason, one of the first calibration steps foreseen to properly balance these reactions obtained a good match in the tailpipe $CO/CO_2$ concentration.

Figure 6a,b report the results for the full engine load, clearly indicating an adequate response of the model on carbon oxide conversion. It is possible to notice that the CO concentration was slightly underestimated around stoichiometric conditions and was overpredicted, as the mixture tended to have very rich values. Such behavior, correspondingly affecting $CO_2$ conversion, was obtained in all the operating conditions, even at partial engine load. Nonetheless, it has to be specified that the discrepancies were contained, in all the cases, within a range of 0.2%, giving rise to a reasonably consistent agreement between numerical and experimental results.

### 5.2. Nitrogen Oxides

Nitrogen oxide reduction occurs if adequate concentrations of CO and $H_2$ are achieved inside the catalyst. The global reactions describing this process in the TWC are:

$$CO + NO \rightarrow CO_2 + \frac{1}{2}N_2$$

$$H_2 + NO \rightarrow H_2O + \frac{1}{2}N_2$$

$$H_2 + 2NO \rightarrow H_2O + N_2O$$

$$CO + 2NO \rightarrow CO_2 + N_2$$

As displayed in Figures 6c, 7c and 8c, at all engine load values, nitrogen oxide conversion efficiency quickly dropped as soon as λ exceeded unity. As the mixture approached even slightly lean conditions, TWC conversion efficiency was approximately zero, and NO tailpipe concentration remained identical to engine-out values. The predicted values suitably reproduced the detected behavior, with some minor inaccuracies at AFR values in the range between 1.01 and 1.04. Taking into account the extreme sensitivity of the converter with respect to these species' concentrations and the related experimental uncertainties, possible improvements to the model could be achieved to collect a higher number of experimental data within the indicated λ range.

Certainly, as mentioned, hydrogen participation in the reaction mechanism should be further investigated because the assumption of Figure 3 may not be respected in all the temperature ranges, especially in specific operating conditions, like dynamic ones. In spite of these considerations, the kinetic scheme, firstly calibrated at full engine load conditions, generally provided a good numerical/experimental agreement even at partial loads.

Finally, around λ ≈ 0.94–0.96, measured emissions showed a small $NO_x$ spike of a few ppm. The numerical model was not capable of capturing such phenomenon, which should be better clarified through additional investigations.

*5.3. Methane*

Methane conversion involves use of the following reactions:

$$CH_4 + 2O_2 \rightarrow 3CO_2 + 2H_2O$$

$$CH_4 + H_2O \rightarrow CO + 3H_2$$

The calibration of these reactions, always carried out at full load and with a continuous check of their influences on the previous ones, allowed to reach an adequate response of the model in the medium-/high-load conditions.

Looking at the measured trends in Figures 6d, 7d and 8d (red and green curves), it is possible to notice that, for all the engine load levels, methane was fully converted only in stoichiometric conditions. As known, the different states of the catalyst surface under lean and rich conditions affects methane conversion in the reactor. In the traditional TWC used for gasoline engines, methane is completely converted in lean conditions. In fact, its diffusion is the limiting step, and hydrogen species, which form on the surface as a result of methane dissociation, are oxidized to $CO_2$ and $H_2O$ by residual oxygen [18]. On the contrary, for the present CNG engine, an efficiency loss in lean conditions was observable, more significantly decreasing the engine load. One of the main reasons could be related to a different response of the Pd storage reactions on the catalyst surface. In order to better understand this phenomenon, dedicated tests could be useful with a linear increase in temperature and constant methane concentration, possibly with the use of a synthetic gas bench (SGB). The model was not able to predict the loss of efficiency that occurs in extremely lean conditions, especially at partial loads where the contribution of water vapor is directly proportional to the temperature decrease. This effect could also be due to an excessive TWC aging in addition to a high presence of $CH_4$ at the exhaust compared to gasoline engines [19].

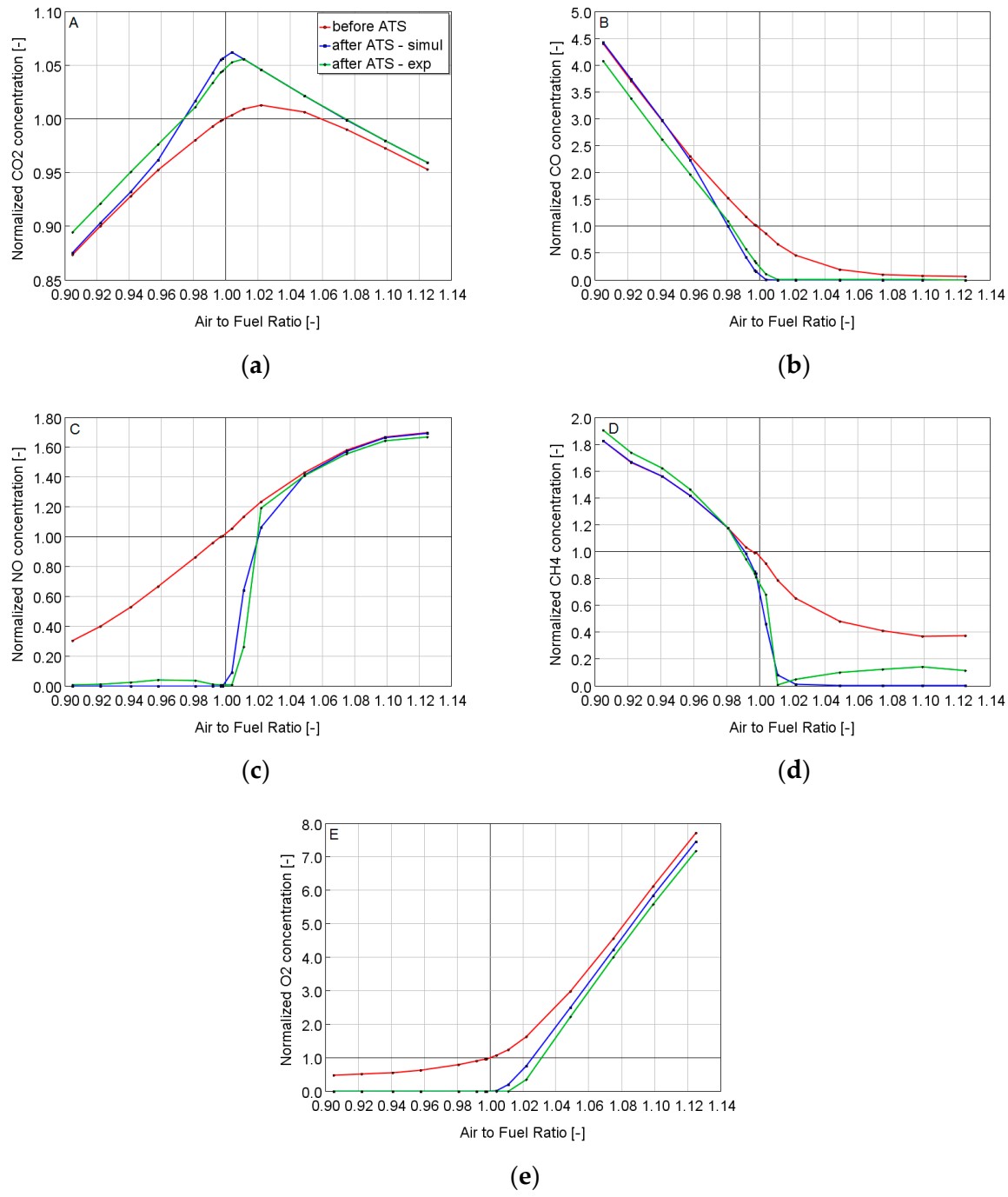

**Figure 6.** SS tests at 100% Load. Gas Temp. $\approx 760\,^\circ$C. (1900 RPM,100% load). (**a**) Num/Exp. $CO_2$ engine out and tailpipe concentration. (**b**) Num/Exp. CO engine out and tailpipe concentration. (**c**) Num/Exp. NO engine out and tailpipe concentration (**d**) Num/Exp. $CH_4$ engine out and tailpipe concentration. (**e**) Num/Exp. $O_2$ engine out and tailpipe concentration.

Moving to tests at AFR lower than unity, the catalyst turned out to be completely unable to reduce methane concentration, with zero conversion efficiency. In these rich conditions, oxygen was the limiting reactant; its concentration was near to zero, and the surface was covered by partial oxidation species, CO and H in particular. The TWC model substantially reproduced this trend at all the engine load values, with a loss of accuracy at an engine load of 40%, as visible in Figure 8.

To sum up, a reasonable agreement between measured and calculated trends was reached, with some discrepancies around stoichiometric and lean AFR values at partial load (Figure 7; Figure 8), indicating the necessity to improve its accuracy in such operating conditions.

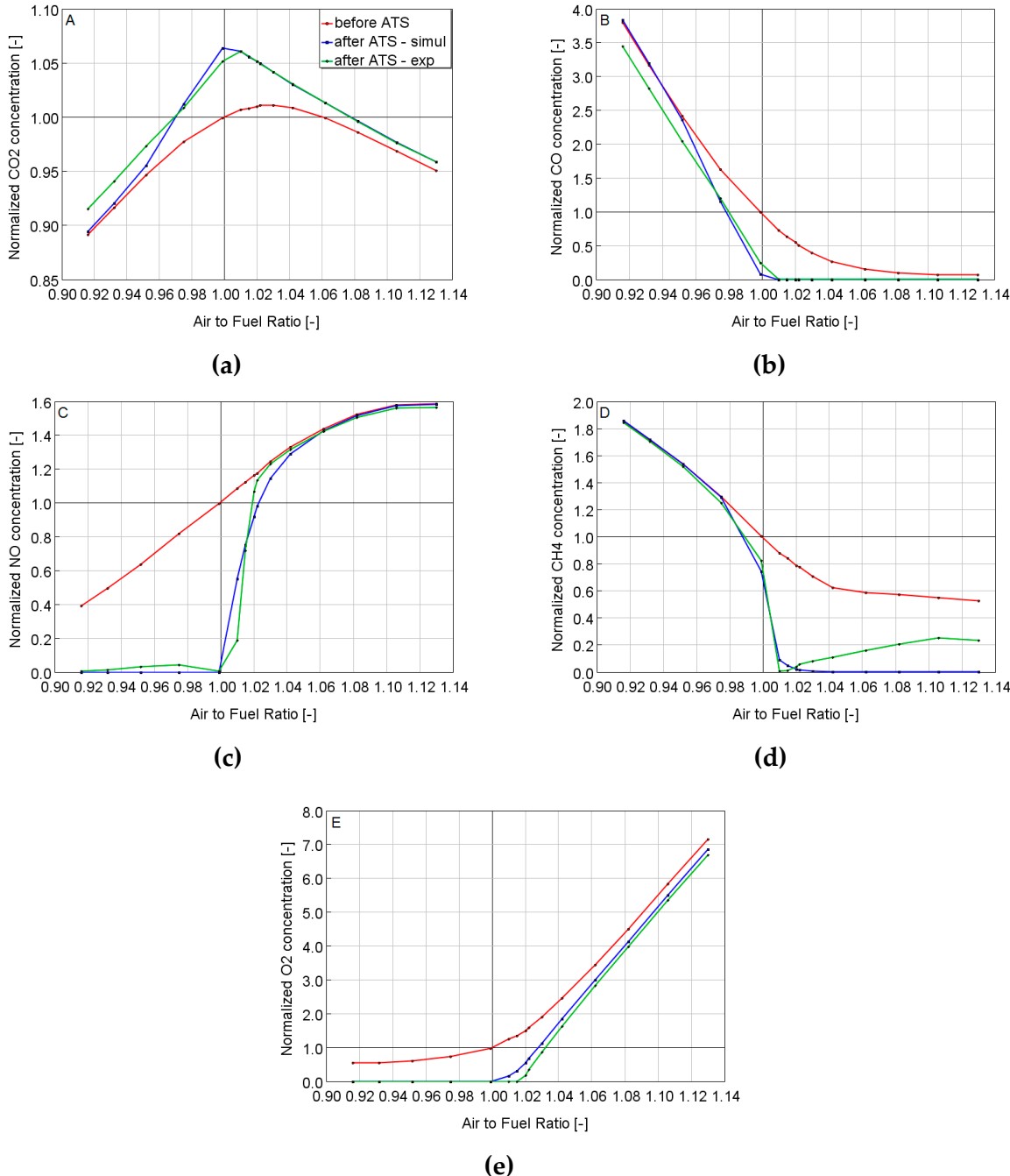

**Figure 7.** SS tests at 80% Load. Gas Temp. ≈ 750 °C. (1900 RPM, 80% load). (**a**) Num/Exp. $CO_2$ engine out and tailpipe concentration. (**b**) Num/Exp. CO engine out and tailpipe concentration. (**c**) Num/Exp. NO engine out and tailpipe concentration (**d**) Num/Exp. $CH_4$ engine out and tailpipe concentration. (**e**) Num/Exp. $O_2$ engine out and tailpipe concentration

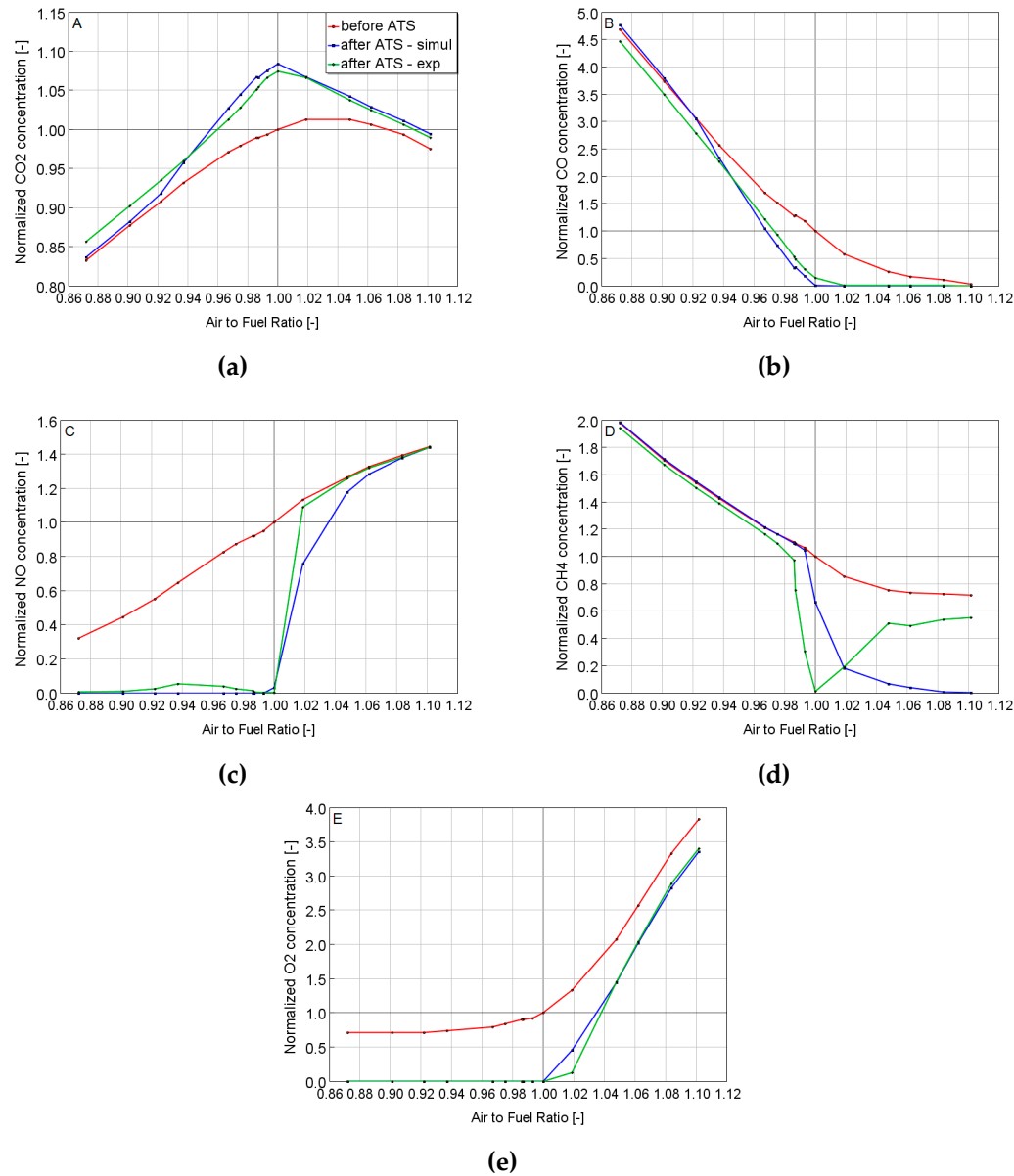

**Figure 8.** SS tests at 40% Load. Gas Temp. ≈ 620 °C. (1900 RPM, 40% load). (**a**) Num/Exp. $CO_2$ engine out and tailpipe concentration. (**b**) Num/Exp. CO engine out and tailpipe concentration. (**c**) Num/Exp. NO engine out and tailpipe concentration (**d**) Num/Exp. $CH_4$ engine out and tailpipe concentration. (**e**) Num/Exp. $O_2$ engine out and tailpipe concentration

## 6. Transient Results

As mentioned, the oxygen storage phenomenon consisted in the formation of different cerium oxides, which affected local concentration values of the main pollutants and significantly modified local AFR values in lean to rich mixture transitions, and vice versa. Conversion of each analyzed species took place in variable times, depending on the concentration and temperature (as in steady-state cases), but also on the aforementioned cerium oxides, which reacted simultaneously with all the other species. It should be emphasized that also in this case, the value of AFR was not exactly what was designated by the ECU control. In fact, there are situations, further described later, in which the catalyst did not exhibit maximum conversion efficiency even though it nominally operated at stoichiometric conditions. It is important to recall that, given the extreme sensitivity of the catalyst to the inlet gas composition, the uncertainty in the knowledge of the actual instantaneous value of lambda represents an open critical issue. In fact, the measured lambda temporal evolution, to be used as a reference to

validate numerical results, can display considerable variability according to the adopted measurement technique. To illustrate this point, Figure 9 (top) reports the AFR profile obtained by means of the Smart $NO_x$ sensor, compared to the imposed target upstream from the TWC. It is possible to notice that the phasing of the measured profiles was correctly reproduced, but the detected maximum and minimum lambda values displayed a discrepancy with respect to the imposed target values (required to be equal to 1.10 and 0.90 respectively). Such discrepancy increased when decreasing the engine load levels. Figure 9b also shows the comparison, for the full engine load case, of the lambda profile detected upstream and downstream from the TWC. Thanks to its fast measurement dynamics, oxygen storage phenomena were captured by the Smart $NO_x$ sensor, with storage and release phases determining the observed differences in the two lambda profiles. On the other hand, the reduced sensor accuracy at the leanest and richest phases, shown in Figure 9, did not permit to fully rely on these data for these transient conditions.

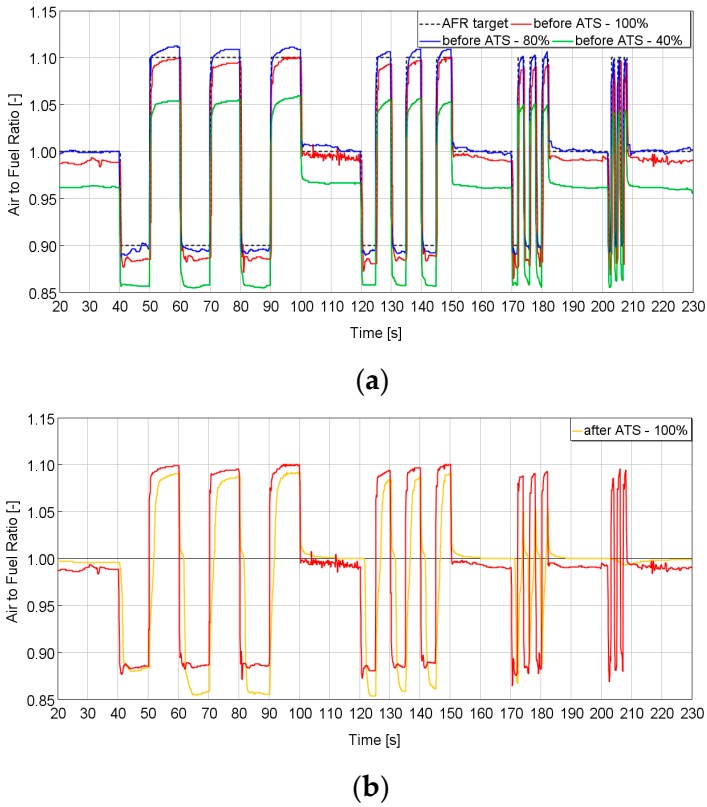

**Figure 9.** Lambda profiles measured by Smart $NO_x$ Sensor upstream from the TWC during dynamic sweeps at different engine load values (**a**). Comparison between measurements upstream and downstream the TWC for the full load case (**b**).

To sum up, when looking at the results reported in the following paragraphs, it should be taken into account that the AFR dynamics at the catalyst inlet were known within the limits of the discussed uncertainties, with the subsequent impact on the initial conditions for the simulations. The following graphs were made with the same logic used in the previous chapter, and the reference values used to normalize the emissions were the same of the respective cases at an equal torque in steady-state conditions. As for the steady-state conditions, a submodel of the reactions describing the oxygen storage was calibrated at high load, and additional tests were then performed at lower loads. Figure 10 reports $CO_2$, $CO$, $NO_x$, and $CH_4$ concentration histories measured upstream and downstream from the TWC, compared with the corresponding calculated profiles at full engine load. Subsequent Figure 11; Figure 12 display the same curves measured and predicted for engine load values equal to 80% and 40%, respectively.

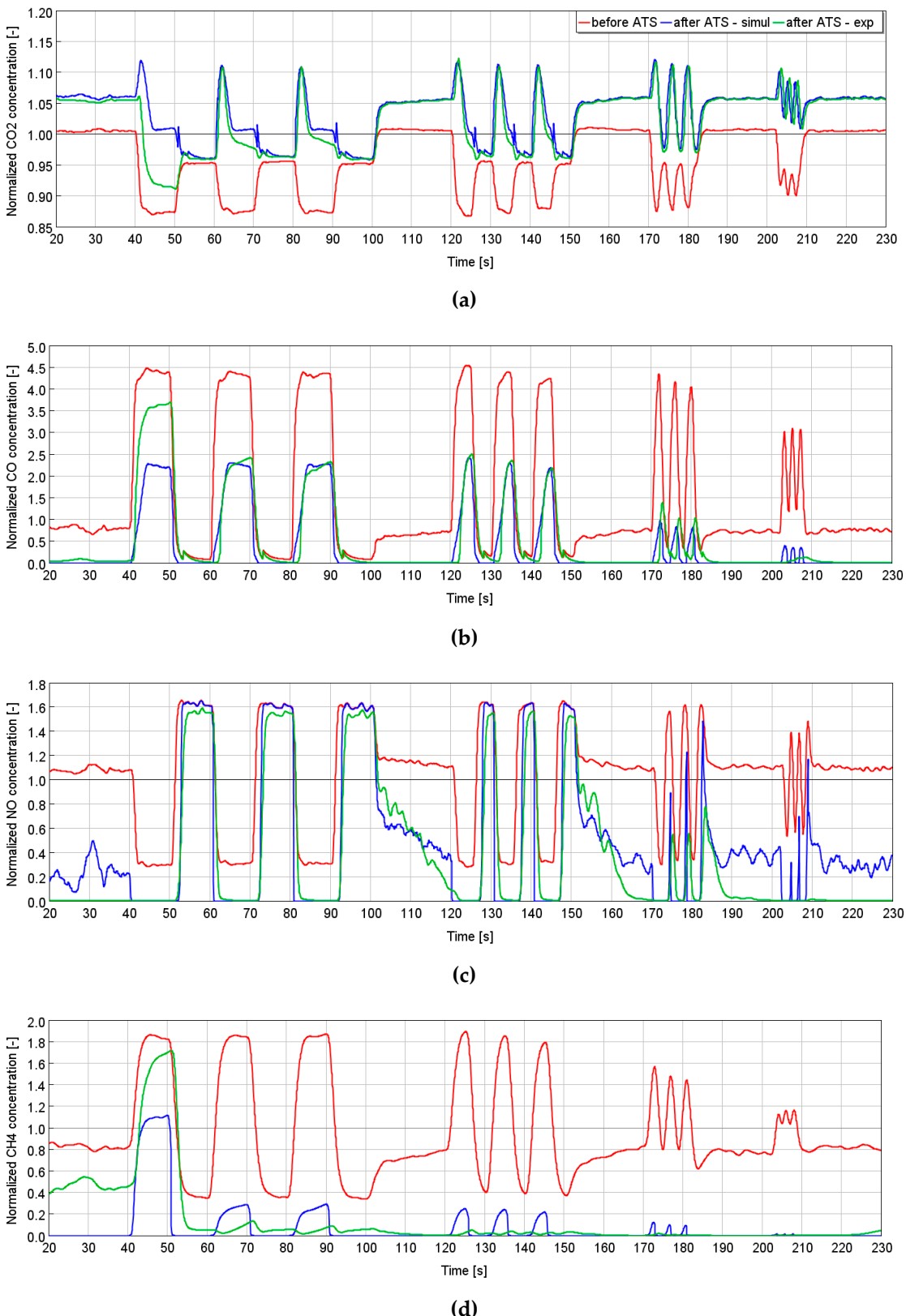

**Figure 10.** Transient conditions at 100% Load. Gas Temperature ≈ 750 °C (**a**) Num/Exp. $CO_2$ engine out and tailpipe concentration. (**b**) Num/Exp. CO engine out and tailpipe concentration. (**c**) Num/Exp. NO engine out and tailpipe concentration (**d**) Num/Exp. $CH_4$ engine out and tailpipe concentration.

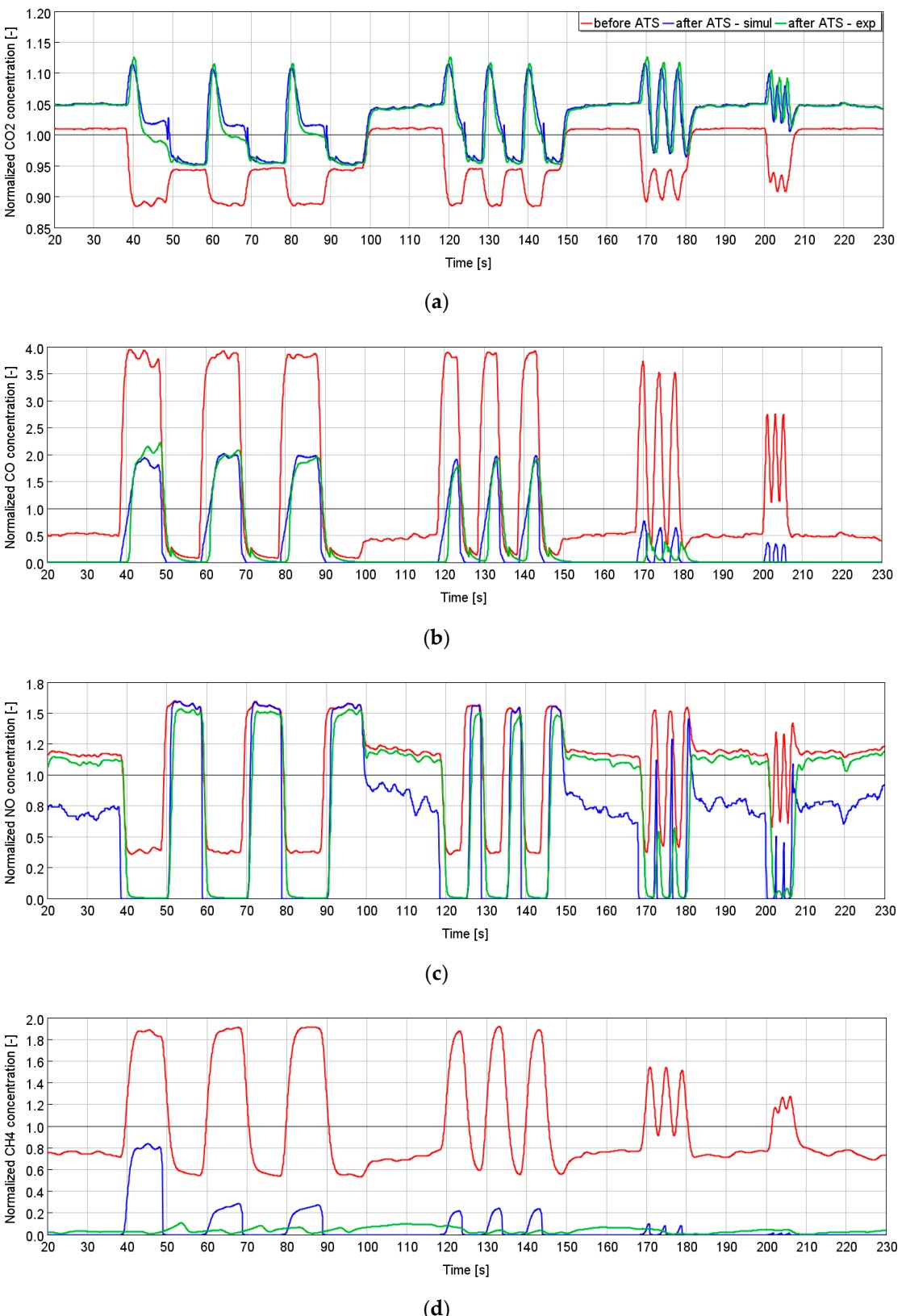

**Figure 11.** Transient conditions at 80% Load. Gas Temperature ≈ 740 °C (**a**) Num/Exp. $CO_2$ engine out and tailpipe concentration. (**b**) Num/Exp. CO engine out and tailpipe concentration. (**c**) Num/Exp. NO engine out and tailpipe concentration (**d**) Num/Exp. $CH_4$ engine out and tailpipe concentration.

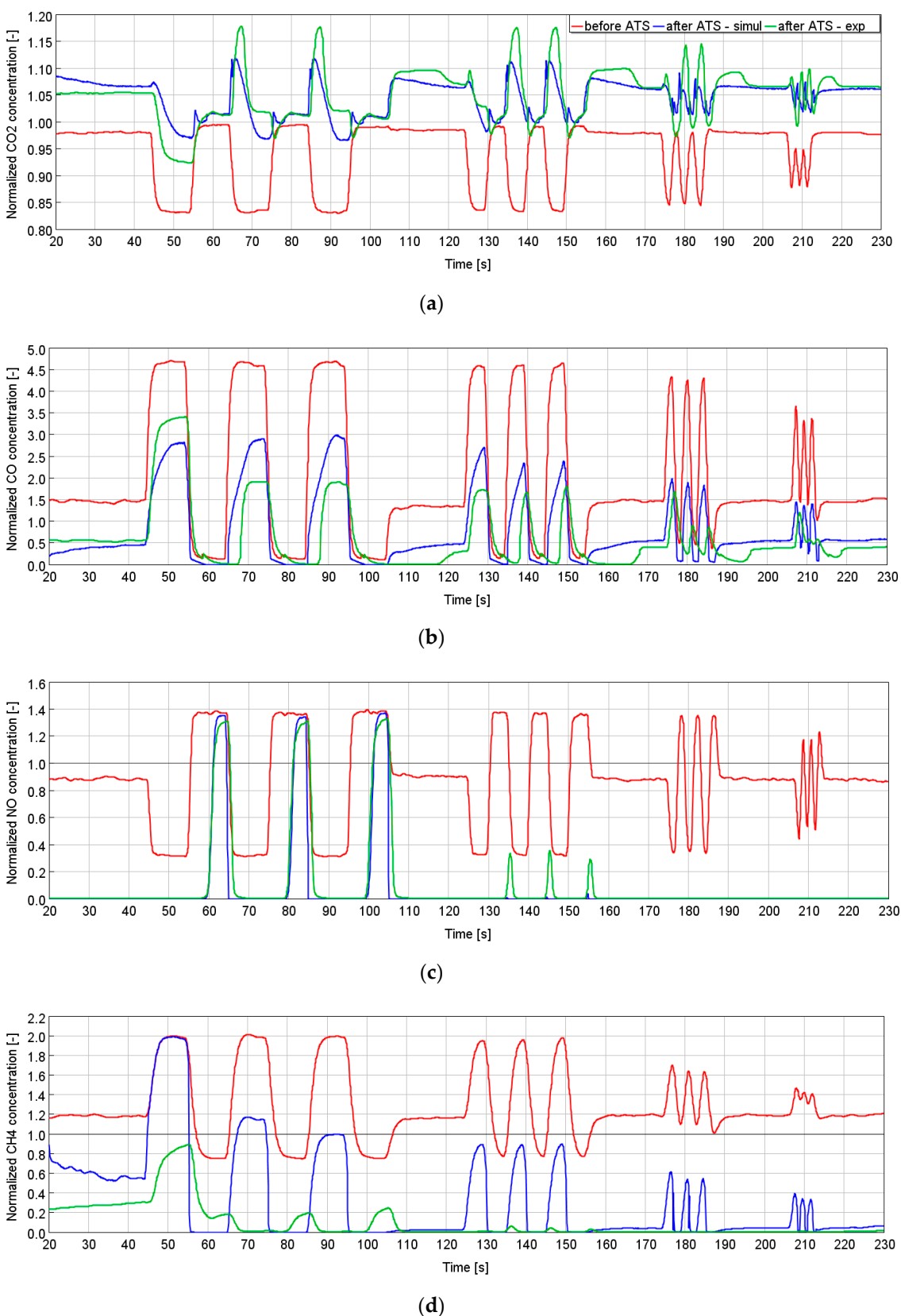

**Figure 12.** Transient conditions at 40% Load. Gas Temperature ≈ 620 °C (**a**) Num/Exp. $CO_2$ engine out and tailpipe concentration. (**b**) Num/Exp. CO engine out and tailpipe concentration. (**c**) Num/Exp. NO engine out and tailpipe concentration (**d**) Num/Exp. $CH_4$ engine out and tailpipe concentration.

### 6.1. Carbon Oxides

Generally speaking, as regards the measured profiles, it is worth specifying that the first AFR step of each sequence of 3 steps (rich to lean) should be considered as TWC pre-conditioning cycles, and should not considered for results repeatability. However, in the subsequent two steps, conversion of the species can be sufficiently characterized. After the lean-to-rich transitions, there were different phases that characterized the oxidation of carbon monoxide. In the 10 s steps, the first part lasted 2–3 s for full load conditions and 4–5 s for medium-low load conditions, where the TWC converted all CO into $CO_2$, with a consequent concentration peak clearly visible in Figures 10a, 11a and 12a. After this initial phase, the conversion efficiency of the TWC was approximately 50%.

The main reason for this change in CO oxidation was due to the fact that, after lean mixture treatment, the TWC was crossed by a higher concentration of $O_2$ that was stored in the catalyst in the form of $CeO_2$. Thus, given the abundant availability of this species, CO oxidation via reaction 19 prevailed with respect to reaction 20 for a certain time interval, which can be suitably calibrated by acting on the parameters that involved these reactions. At high load, as shown in Figure 10; Figure 11 at 100% and 80% torque levels, the model was able to predict very accurately the dynamics involving the conversion of CO and $CO_2$ (with the exception of the first step).

It is interesting to note that there was a sort of adjustment on this efficiency value in the 10–5 s amplitude transitions, demonstrating the achieved equilibrium conditions of the system after the first phase with maximum conversion efficiency. In the 2 s transitions, instead, the timing was not sufficient to ensure that the system returned to balance. Thus, the change in behavior of the catalyst and, therefore, the change in the rate of reactions involving carbon oxides were interrupted, with a conversion efficiency of approximately 80%. Experimentally, the transitions of 1 Hz were too fast to verify the influence of the AFR sweeps on the high conversion efficiency due to dilution effects along the analyzer's measurement line, which was gradually greater as the load decreased. Finally, it is interesting to notice that the model was also able to capture the variation of the CO profile slope when the engine load was reduced, reproducing the steep transitions of full load and the slower dynamics at partial load.

### 6.2. Nitrogen Oxides

Looking at Figures 10c, 11c and 12c, as described for carbon oxides, after the lean-to-rich transition there was a first phase in which the conversion efficiency was maximum, while in a subsequent phase there was no impact on the reduction of nitrogen oxides. As said, with respect to the starting model included in the simulation software, the addition of reaction 22 allowed a more immediate calibration, since it permitted to act directly on the NO species, which would otherwise be managed exclusively by an indirect calibration on carbon oxide reactions.

Generally speaking, the obtained model accurately reproduced the conversion within the catalyst during these transient phases at all the considered engine load levels. Some minor discrepancies could be observed, similar to what occurred in the steady-state conditions (see Figures 6c, 7c and 8c), around the stoichiometric λ value, giving rise, as an example, to a slightly slower NO decrement between 100–120 s.

### 6.3. Methane

As already discussed, in all the considered steady-state conditions, methane was fully converted only at the stoichiometric condition, while an efficiency loss was observed in lean cases, and no conversion occurred in rich ones. On the contrary, as visible in Figure 10, during the dynamic λ transitions, the $CH_4$ tailpipe concentration displayed a different behavior, TWC shows a high and constant conversion efficiency even in rich conditions. The only exception in Figure 10 was represented by the first spike, but, as already mentioned, the first step could be neglected as TWC pre-conditioning. A similar response was obtained in the experiments at lower engine loads reported in Figures 11

and 12, in which methane concentration profiles at the catalyst outlet appeared nearly unaffected by the inlet concentration dynamics. In order to try to capture the observed TWC behaviors, the reaction mechanism was modified, including some steps involving palladium. In fact, it is known that methane can interact with the oxides of noble metals, such as palladium [18]. For this reason, the following two reactions have been added to the kinetic scheme, identified in the Appendix A with the numbers 23 and 24.

$$Pd + O_2 \rightarrow PdO_2$$
$$PdO_2 + CO \rightarrow Pd + 2CO_2$$

The link between methane and palladium oxide was created by modifying the expression rate of the methane oxidation reaction (reaction 5 in the Appendix A), introducing an inhibition term $\Omega$, defined as:

$$\Omega = \frac{mol\ of\ PdO_2}{2 \cdot mol\ of\ Pd + mol\ of\ PdO_2}. \tag{6}$$

The kinetic parameters of the added/modified reactions were only calibrated at full engine load, then applied to the other test cases. As visible in Figure 10d, thanks to this modification, the model could reproduce a consistent methane conversion during the AFR sweep, especially in rich phases, despite not being completely abated as shown in the measured profile. At lower engine load values (Figure 11; Figure 12d), the agreement between experimental and predicted methane histories was less satisfying, but it is likely that further calibration work, possibly combined with additional experiments, could significantly improve these results.

## 7. Conclusions

A predictive model of catalyst behavior with oxygen storage and release of a heavy-duty engine powered with NG was developed. The need to readapt the existing reaction scheme for TWC of gasoline-powered engines in order to make them suitable for NG has been demonstrated. A protocol for the calibration of the main reactions was identified through an iterative trial-and-error approach. Calibration of the reactions kinetic scheme was specifically carried out starting from the emissions at full engine load during a wide AFR sweep, both in steady-state and transient conditions. It represents a good starting point and a first goal in the use of a simplified scheme to manage the complex phenomena occurring in a TWC for the aftertreatment of emissions by NG engines. Indeed, the response of the model in terms of emissions was adequate in high-load conditions and was still acceptable under medium-low load conditions, considering also the higher measurement uncertainties from the analyzer devices present in these circumstances. $CH_4$ oxidation represents the major open point of the current scheme in lean conditions and during the dynamic lambda scan, characterized by different phenomena with respect to similar stationary conditions. Generally speaking, a reasonable predictivity of the model was obtained, resulting in a sufficiently adequate representation of the catalyst reactivity in dynamic conditions. Cold-start phases before the light-off temperature and the analysis in driving cycles representative of the real transient ATS working conditions, such as the WHTC, need further investigation as well as model validation with different NG engines.

**Author Contributions:** Conceptualization, C.B. and D.D.M.; methodology, C.B., S.G., P.N. and F.G.R.; validation, D.D.M., C.B. and V.F.; investigation, D.D.M., C.B. and V.F.; writing—original draft preparation, D.D.M.; writing—review and editing, D.D.M. and V.F.; supervision, C.B., V.F., P.N. and S.G.; project administration, C.B.; funding acquisition, C.B.

**Funding:** This research received no external funding.

**Acknowledgments:** The authors would like to thank FPT industrial for the right to disclose and publish these excerpts of the overall research activity. Furthermore, authors would like to thank Dimitrios Tsinoglou from FPT Industrial for his scientific assistance in the kinetic scheme calibration.

**Conflicts of Interest:** The authors declare no conflict of interest.

## Definitions/Abbreviations

| | |
|---|---|
| AFR | Air-to-fuel ratio |
| ATS | Aftertreatment system |
| CI | Compression ignition |
| CLD | Chemi-luminescence detector |
| CNG | Compressed natural gas |
| ECU | Engine control unit |
| FID | Flame ionization detector |
| IRD | Infrared detector |
| NG | Natural gas |
| NMHC | Non-methane hydrocarbons |
| OEM | Original equipment manufacturer |
| OSC | Oxygen storage capacity |
| PFI | Port fuel injection |
| PGM | Platinum group metals |
| PMD | Paramagnetic detector |
| SI | Spark ignition |
| SS | Steady state |
| SGB | Synthetic gas bench |
| SNS | Smart $NO_x$ Sensor |
| THC | Total hydrocarbons |
| TWC | Three-way catalyst |
| WGS | Water–gas shift |
| ZFAS-U | Universal lambda sensor |

## Appendix A

Chemical reactions included in the final mechanism:

| | |
|---|---|
| 1 | $CO + \frac{1}{2}O_2 \rightarrow CO_2$ |
| 2 | $H_2 + \frac{1}{2}O_2 \rightarrow H_2O$ |
| 3 | $C_3H_8 + 5O_2 \rightarrow 3CO_2 + 4H_2O$ |
| 4 | $C_3H_6 + \frac{9}{2}O_2 \rightarrow 3CO_2 + 3H_2O$ |
| 5 | $CH_4 + 2O_2 \rightarrow 3CO_2 + 2H_2O$ |
| 6 | $CO + H_2O \rightarrow CO_2 + H_2$ |
| 7 | $C_3H_8 + 3H_2O \rightarrow 3CO + 7H_2$ |
| 8 | $C_3H_6 + 3H_2O \rightarrow 3CO + 6H_2$ |
| 9 | $CH_4 + H_2O \rightarrow CO + 3H_2$ |
| 10 | $CO + NO \rightarrow CO_2 + \frac{1}{2}N_2$ |
| 11 | $H_2 + NO \rightarrow H_2O + \frac{1}{2}N_2$ |
| 12 | $C_3H_6 + 9NO \rightarrow 3H_2O + 3CO_2 + \frac{9}{2}N_2$ |
| 13 | $H_2 + 2NO \rightarrow H_2O + N_2O$ |
| 14 | $N_2O + H_2 \rightarrow H_2O + N_2$ |
| 15 | $CO + 2NO \rightarrow CO_2 + N_2O$ |
| 16 | $N_2O + CO \rightarrow CO_2 + N_2$ |
| 17 | $H_2 + 2CeO_2 \rightarrow H_2O + Ce_2O_3$ |
| 18 | $H_2O + Ce_2O_3 \rightarrow H_2 + 2CeO_2$ |
| 19 | $CO + 2CeO_2 \rightarrow CO_2 + Ce_2O_3$ |
| 20 | $CO_2 + Ce_2O_3 \rightarrow CO + 2CeO_2$ |
| 21 | $\frac{1}{2}O_2 + Ce_2O_3 \rightarrow 2CeO_2$ |
| 22 | $NO + Ce_2O_3 \rightarrow \frac{1}{2}N_2 + 2CeO_2$ |
| 23 | $Pd + O_2 \rightarrow PdO_2$ |
| 24 | $PdO_2 + CO \rightarrow Pd + 2CO_2$ |

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
