# Peer review of "Modeling of Three-Way Catalyst Dynamics for a Compressed Natural Gas Engine during Lean–Rich Transitions"

_applsci, doi:10.3390/app9214610_

Round 1

Reviewer 1 Report

1) All the abbreviations used through the paper should be in the Abbreviations list and explained through the paper text – for example CNG is missing in the Abbreviations list. Check all the others. Also, many abbreviations should be unified through the paper text – for example CH4 is written sometimes with 4 in the index, sometimes 4 is not in the index. Correct through paper text all the abbreviations.

2) All the equations through the paper text should be numerated and call on each equation should be putted in the paper text.

3) Figure 4 - Empirical correlation is obtained in [17] for a standard gasoline engine. Can it be used in an NG engine? What is the expected error of this polynomial for NG engine?

4) Section 4 – Model Validation – in this section should be included at least governing equations of the “quasi-steady” developed model. Furthermore, in this section should be presented validation results of the model (similar but not the same to ones presented in Sections 5 and 6).

5) Line 229 – what is PGM?

6) In almost all figures which shows the results is used abbreviation ATS. What this abbreviation represents? It is quite hard to understand performed results if this information about ATS cannot be found in the paper text.

7) Line 270 – it is stated that in the results AFR is calculated by R49 formula. Why is not used GT-Suite formula as presented in Figure 2? It will be much easier to present simulation results in relation to already implemented AFR formula in the GT-Suite. Why was important to calculate AFR exactly with R49 formula? It should be commented in the paper text.

8) Line 287 – instead of “Figures 5A-B” it should be written “Figures 5, 6 and 7 (A-B)”.

9) Figure 8 – why in the paper text and in figure title usage of (top) and (bottom) – it should be used (a) and (b).

10) Only the comment: Simulating of catalyst dynamics during transient operation is hard task. The authors perform it very well, and it is given a reasonable guidelines for future research and model improvement. Any numerical modeling of internal combustion engine (or any of its parts) during transient operation is hard and complex task – which is in this case well performed.

Final Reviewer remark: This is interesting, well performed and useful paper which will be interesting to a wider scientific community. However, it requires MINOR REVISION, correction of some obvious mistakes and abbreviations as well as significantly improving of model validation (and presenting at least governing equations of the developed model). After corrections, this paper will have my recommendation for publication in Applied Sciences.

Author Response

The authors thank the reviewer for his kind comments.

Response to Reviewer 1 Comments

Point 1: All the abbreviations used through the paper should be in the Abbreviations list and explained through the paper text – for example CNG is missing in the Abbreviations list. Check all the others. Also, many abbreviations should be unified through the paper text – for example CH4 is written sometimes with 4 in the index, sometimes 4 is not in the index. Correct through paper text all the abbreviations.

Response 1: Thanks for this note. The abbreviation for CH4 has been added and the subscript "4" has been appropriately corrected (introduction). The same was also done for the "x" in NOx (line 35,101,149,153,172,310, ch. “Transient results”).

Point 2: All the equations through the paper text should be numerated and call on each equation should be putted in the paper text.

Response 2: Three equations have been numbered (in red).

Point 3: Figure 4 - Empirical correlation is obtained in [17] for a standard gasoline engine. Can it be used in an NG engine? What is the expected error of this polynomial for NG engine?

Response 3: Actually, hydrogen concentration of hydrogen present in a CNG-engine exhaust is not known. To our knowledge, moreover, in literature there is no other indication. Therefore, in order to still evaluate the contribution that hydrogen has in the conversion of the various species, this report was used which could present discrepancies that can only be evaluated by comparing the measurements performed at ETB (Engine Test Bench) with SGB (Sinthetic Gas Bench).

Point 4: Section 4 – Model Validation – in this section should be included at least governing equations of the “quasi-steady” developed model. Furthermore, in this section should be presented validation results of the model (similar but not the same to ones presented in Sections 5 and 6).

Response 4: Governing equations that describe the model are included in the manual and are reported in the reference [17]. In order to appropriately appreciate the calibration activity of the kinetic scheme, figure 5 has been added which shows the wide deviation that occurs in the conversion of CO and CH4 using the uncalibrated model, already implemented in GT-Suite.

Point 5: Line 229 – what is PGM?

Response 5: PGM is the abbreviation of Platinum Group Metals.

6) In almost all figures which shows the results is used abbreviation ATS. What this abbreviation represents? It is quite hard to understand performed results if this information about ATS cannot be found in the paper text.

Response 6: Thanks for this comment. Actually in the text we talked about Three-Way Catalyst while in the legend of the figures is indicated Aftertreatment System (ATS).

7) Line 270 – it is stated that in the results AFR is calculated by R49 formula. Why is not used GT-Suite formula as presented in Figure 2? It will be much easier to present simulation results in relation to already implemented AFR formula in the GT-Suite. Why was important to calculate AFR exactly with R49 formula? It should be commented in the paper text.

Response 7: The lambda calculated by GT-Suite is a function of the chemical species measurement in exhaust evaluated by the analyzers, that are an input data of the model. Nonetheless, the calculation of the lambda represented by R49 formula is very close to what was evaluated by GT-Suite (Fig. 2), especially around the stoichiometric. A deviation of about 1% is visible only in extremely rich or lean conditions, due to greater uncertainties on the chemical species (oxygen concentration, hydrogen etc.). Since the target of this research activity is to evaluate the conversion of the main pollutants in a wide lambda range, it was decided to represent each test at fixed lambda in Steady-State, calculated in the same way through the R49 formula. In this way these slight uncertainties are neglected and in the graphs of chapter 5 is thus better appreciated, focusing on a single parameter, the conversion efficiency of the main species and the goodness of the kinetic scheme.
A brief comment has also been added to the text.

8) Line 287 – instead of “Figures 5A-B” it should be written “Figures 5, 6 and 7 (A-B)”.

Response 8: Updated in the text.

 9) Figure 8 – why in the paper text and in figure title usage of (top) and (bottom) – it should be used (a) and (b).

Response 9: Updated in the text.

10) Only the comment: Simulating of catalyst dynamics during transient operation is hard task. The authors perform it very well, and it is given a reasonable guidelines for future research and model improvement. Any numerical modeling of internal combustion engine (or any of its parts) during transient operation is hard and complex task – which is in this case well performed.

Response 10: Thank you very much for your comments. His considerations were highly appreciated.

Reviewer 2 Report

This is an interesting paper. Some comments:

Line 75: It reads that a CI engine is used. So it is a dual fuel engine? How big is the diesel share (in terms of energy)?

Line 83 - 85: This sentence does not seem to fit in the context at this place.

Line 123 - 125: The indices should be written as such, like in the equation.

Line 274: Reference to Fig. 4 is unclear. There is no A to E.

Page 12, last but one line above Fig. 8: Number of Figure is missing.

Author Response

The authors thank the reviewer for his kind comments.

Response to Reviewer 2 Comments

Point 1: It reads that a CI engine is used. So it is a dual fuel engine? How big is the diesel share (in terms of energy)?

Response 1: Thanks for this note. I preferred to eliminate this wording because, this engine was originally a diesel-engine, and from it was derived an engine powered only with Natural Gas. It is not a Dual Fuel. In the Table 1 you can find the Rated Power. For confidentiality reasons, no further data has been highlighted.

Point 2: Line 83 - 85: This sentence does not seem to fit in the context at this place.

Response 2: Updated in the text.

Point 3: Line 123 - 125: The indices should be written as such, like in the equation.

Response 3: Thanks for this note. Updated in the text.

Point 4: Line 274: Reference to Fig. 4 is unclear. There is no A to E.

 Response 4: Thanks for this note. It was a mistake. Updated in the text.

Point 5: Page 12, last but one line above Fig. 8: Number of Figure is missing.

Response 5: Updated in the text.

Reviewer 3 Report

See the attached.

Author Response

The authors thank the reviewer for his kind comments.

Response to Reviewer 3 Comments

Point 1: Line 0. Always capitalize the first letter of the words of the article title, as shown in red above.

Response 1: Thanks for this note. Updated in the text.

Point 2: Line 40. Should read: ..which acts as a stabilizer and a medium for oxygen storage ….

Response 2: Thanks for this correction. Updated in the text.

Point 3: Should read: … The results’ accuracy …..

Response 3: Updated in the text.

Point 4: Line 82. The label for the Table 1 should be above the Table. Box the Table and make the Table presentable.

Response 4: Updated in the text and reformatted table.

Point 5: Line 91. Ditto for Table 2. Same comment as for line 82 applies.

Response 5: Updated in the text and reformatted table.

Point 6: Should read …….. suffers higher errors.. In short rephrase the sentence with better choice of words.

Response 6: Modified in "it presents major uncertainties".

Point 7: Line 120. Write the equation with font size 9 or 10. Leave a line space above and below the equation. Label this as Equation 1 and label subsequent equations. Reference all sources where your equations are derived from.

Response 7: Labels have been added to identify each equation and references are available.

Point 8: Line 131. Rephrase the sentence: …..permits to neglect….

Response 8: Modified in “allows to disregard the use of sensors and their mentioned inaccuracies”

Point 9: Lines 188. Rephrase this for clarity …..while vary during dynamic λ sweep tests.

Response 9: Modified in “Their values can be considered constant because these tests are carried out at fixed load and engine speed. Very slight fluctuations in the measurement are to be attributed to the sensor acquisition dynamics.”

Point 10: Line 200. Use black color for this Picture legend. Figure 4. ?2/??

Response 10: Thanks for this correction. Updated in the text.

Point 11: Lines 223-225. Remove the equation from the lines and set it up as an equation. Do not put an equation in a line of a sentence.

Response 11: Updated in the text.

Point 12: Line 229. Remove the line between the two words. ….implemented. PGM chemical

Response 12: Updated in the text.

Point 13: Line 271. Should read …Looking at the x-axis of Figures 5,6,7,. Use comma instead of dash

Response 13: Updated in the text.

Point 14: Line 292. In Line 292, you wrote… within a range of 0.2%, …. On line 206, you wrote : …by the analyzer, is approximately equal to 0,2% even… Let’s agree that we will use (.) instead of (,) to separate decimal points throughout the entire document.

Response 14: Thanks for this kind note. Updated in the text.

Point 15: Page 9. Line 7 from below the last paragraph. Rephrase sentence. It lacks clarity. …which resulted equal to the inlet one

Response 15: Modified in “with zero conversion efficiency”.

Point 16: Page 13. Figure 8, Leave a line space between the figure and the last text line.

Response 16: Updated in the text.

Point 17: Generally, if your label on the pictures were 5d, 6d and 7d. Use same in the text and not 5D, 6D and 7D. This applies to all labels of pictures within the texts. Review and re-label them.

Response 17: Updated in the text.

Point 18: You did not derive all the equations used here. You should reference sources.

Response 18: The refencence used for each equation has been highlighted, where missing.

Reviewer 4 Report

A predicted model of catalyst behavior with oxygen storage and release of a HD engine powered with CNG has been described in this paper with clear pictures and explanations. My congratulations to the authors for clarify this matter. 

Author Response

The authors thank the reviewer for his kind comments.